# Development of Evaluation Methods for Anti-Glycation Activity and Functional Ingredients Contained in Coriander and Fennel Seeds

Akiyoshi Sawabe [1,2,*,†], Atsuyuki Yamashita [2,†], Mei Fujimatsu [1] and Ryuji Takeda [3]

1   Department of Applied Biological Chemistry, Faculty of Agriculture, Kindai University, 3327-204 Nakamachi, Nara 631-8505, Japan; mei12070613@gmail.com

2   Graduate School of Agriculture, Kindai University, 3327-204 Nakamachi, Nara 631-8505, Japan; a.yamashita31@gmail.com

3   Department of Nutritional Sciences for Well-being, Faculty of Health Sciences for Welfare, Kansai University of Welfare Sciences, 3-11-1 Asahigaoka, Kashiwara 582-0026, Japan; rtakeda@tamateyama.ac.jp

*   Correspondence: sawabe@nara.kindai.ac.jp

†   These authors contributed equally to this work.

**Abstract:** Spices are known to have various physiological functions. We focused on the anti-glycation effects of spices, researched anti-glycation active ingredients in coriander (*Coriandrum sativum* L.) and fennel (*Foeniculum vulgare*) seeds, and conducted experiments using human skin-derived fibroblast TIG-110 cells as a model of glycation. We isolated 11 compounds from two spice seeds and found several substances that showed anti-glycation activity. A new compound (5,5′-diallyl-2,2′-diglucopyranosyl-3,3′-dimethoxy diphenyl ether) was isolated from fennel seeds and showed high anti-glycation activity with an IC50 value of 0.08 mM, thereby indicating a high anti-glycosylation activity. In this study, we established a glyoxal (GO)-induced glycation test method for human skin cells, confirmed the anti-glycation effect of spice seeds using this glycation induction model, and found that the exposure of TIG-110 human skin-derived fibroblast cells to GO reduced cell viability. The most stable conditions for cell viability were found to be a GO concentration of 1.25 mM and a culture time of 48 h. We evaluated extracts and isolates of spice seeds using this model as a model test for glycation induction. We conducted qualitative and quantitative analyses of carboxymethyl lysine (CML), a type of AGE, to determine the relationship between cell viability and AGEs. The relationship between cell viability and the amount of CML was correlated. Establishing a glycation induction model test using skin cells makes it possible to quickly screen extracts of natural ingredients in the future. Moreover, the results of this model showed that extracts of two spice seeds and their isolates have high anti-glycation activity, and they are expected to be used as cosmetics, health foods, and pharmaceutical ingredients.

**Keywords:** spice seed; coriander; fennel; AGEs (advance glycation end-products); anti-aging; evaluation method for anti-glycation activity

## 1. Introduction

Spices serve as seasonings to add flavor, aroma, and color to food. In particular, dried spices, including roots, leaves, and seeds, are used as seasonings in Japan and many parts of the world [1,2]. Although spices come from various origins, many were introduced to Europe via the Silk Road from Central Asia, and from the African continent during the Age of Exploration. In botanical classification, they often belong to the following families: *Perilla*, Solanaceae, Brassicaceae, Seriaceae, Liliaceae, and Gingeraceae; they are similar to the vegetables consumed by humans daily.

Spice seeds have specific characteristics in terms of aroma, taste, and coloration, and the effects they impart during cooking include flavoring, odor correction, zest, and coloration. These effects are collectively called the four primary effects. Essential oils, in

addition to seeds, include aromatic components, and each spice has its unique aromatic and flavor properties. Numerous studies have reported that some essential oil components exhibit high physiological and pharmacological activities, such as antioxidant and antibacterial effects [3]. This study focused on the water-soluble fraction of spice seeds, excluding the essential oil component, which has not yet been reported to have biological activity.

Spices including coriander (*Coriandrum sativum* L.) and fennel (*Foeniculum vulgare*) grow wild in India and Sri Lanka. The warm and humid climate and rich soil conditions in India and Sri Lanka enable the propagation of many plant species in different parts of these countries [4–6]. The bioactivities of these two spices (coriander and fennel) include antioxidant [7] and antimicrobial [8] effects, described previously herein, and previous studies have reported that all the herbs have these activities. In addition to the previously reported bioactivities, we suggest the presence of effects that are beneficial for anti-aging properties and the prevention of lifestyle-related diseases [9,10].

Maillard reactions (glycation reactions) form AGEs (advance glycation end-products), the end products of the non-enzymatic combination of reducing sugars, such as glucose and fructose, with amino acid residues, such as lysine. Glycation reactions in vitro are known to occur over a long period, and products accumulate in various organs, leading to aging and disease [11–13]. A study by Jeanmaire et al. (2001) [14] confirmed AGE formation and progressive accumulation in women over 35 years of age in the abdominal skin dermis. It has also been reported that the formation and accumulation of AGEs in the skin are further accelerated by exposure to solar radiation. The accumulation of AGEs alters fibroblast morphology and induces apoptosis, aging, and cellular damage [15,16]. It also causes appearance changes, such as skin wrinkling and sagging, owing to a decrease in the elasticity of collagen fibers in the dermis. Therefore, the relationship between glycation and skin over time is significant [17].

Spices have been widely used in Southeast Asian countries such as Bangladesh, India, and China [4–6]. In particular, the two species used in this study have diverse flavonoid compounds and have been reported to have many marked antioxidant activities [7–9].

There have been active exploratory studies of anti-glycation components in recent years, yet there have been very few reports of components in spice seeds with anti-glycation activities.

AGEs are formed in vitro by various factors. However, since the pathways for AGE formation are specific, it is essential to inhibit these particular pathways [18]. There are mainly two types of existing anti-glycation activity tests. One method the production of AGEs via a nonspecific pathway by culturing glucose and proteins, termed the primary glycation pathway. The other method is to expose cells to highly-reactive dicarbonyl compounds such as glyoxal (GO) or methylglyoxal, which are glycation intermediates [19], to immediately and specifically produce the AGEs carboxymethyl lysine (CML) and carboxymethylarginine [20–22].

We previously formed AGEs by culturing glucose and bovine serum-derived protein (BSA) in vitro by modifying the method described in previous reports [23–25]. Using this experiment in vitro as a benchmark, we searched for components associated with the anti-glycation activity of the spice seeds extracts.

This study examined model experiments of glycation induction using human skin-derived cells for the isolated compounds. Whereas previous studies have found a model of enhanced glycation in human skin keratinocytes—HaCaT cells [26,27]—there have been very few studies on models of induced glycation using dermal-derived cells. We therefore used GO as a saccharification inducer. Aspects of previous studies have been somewhat ambiguous concerning the concentration and culture time of saccharification inducers such as GO. In this study, we examined the anti-glycation activities of spice seed extracts and isolates after creating a model of induced glycation using human-derived dermal fibroblasts, TIG-110 cells.

## 2. Materials and Methods

### 2.1. General Experimental Procedures

For silica gel column chromatography, glass columns ($\Phi$5 cm $\times$ 90 cm, $\Phi$2.5 cm $\times$ 30 cm) were used and silica gel (C-300 Wako gel, Fujifilm Wako Pure Chemical Co., Tokyo, Japan) was used as a packing material. The eluting solvent was prepared with a mixture of chloroform, methanol, and distilled water in appropriate proportions. Gel filtration was performed using TSK gel HW-40F (Tosoh Co., Ltd., Tokyo, Japan) as packing material in a glass column ($\Phi$50 mm $\times$ 1000 m, $\Phi$60 mm $\times$ 950 mm). Carbonated water (pH 5.0), distilled water (pH 7.0), ammonia water (pH 9.0), and 50% methanol were used in sequence as eluting solvents, with an elution rate of 13–15 mL/min. To detect substances in the eluting solvents, absorbance at 280 nm was measured with a UV meter (UVNICON UV-2800 Advantec Toyo Co., Ltd., Tokyo, Japan). Substances were further fractionated and isolated according to the obtained absorption curve. Using a JEOL JMS-700M Station mass spectrometer (FAB/MS) and an LCMS-2020 mass spectrometer (Shimadzu instrument, Kyoto, Japan), the mass counts of the isolates were measured. Then, nuclear magnetic resonance (NMR) spectra were measured using a BRUKER AVANCETM III Nanobay nuclear magnetic resonance spectrometer (400 MHz) with DMSO-d6 or CD$_3$OD as a solvent.

### 2.2. Plant Material

The coriander (*Coriandrum sativum* L.; CSL) seeds and fennel (*Foeniculum vulgare*; FNF) seeds used in the study were commercially available.

### 2.3. Extraction and Isolation

Coriander seeds (910 g) were chopped by a commercial mixer and homogenized after addition of hot water (16.2 L). Cold methanol (37.8 L) was added to the hot water extract to 70% content. The 70% methanol extract was allowed to stand in the dark for 1 week. Similarly, fennel seeds (1200 g) were extracted with hot water using the same procedure as for coriander seeds and adjusted to 70% using ethanol (37.8 L). Each extract was separated into a filtrate and residue with a reduced-pressure suction filtration system. The filtrate was further concentrated under reduced pressure. The concentrated extracts were extracted with 1-hexane, and then 1-butanol. The two spice extracts obtained from this process, 1-hexane extract (coriander seed: 4.3 g; fennel seed: 31.0 g), 1-butanol extract (coriander seed: 5.7 g; fennel seed: 34.5 g), and the aqueous layer extract were separated into three fractions.

The 1-butanol extract of the coriander seed was separated by silica gel column chromatography using silica gel. The fractions obtained from the 1-butanol extract of coriander seeds were further separated sequentially using silica gel column chromatography or gel filtration. In total, six compounds **1–6** were isolated by separating the coriander seed 1-butanol extract.

The 1-butanol extract of the fennel seed was chromatographed on Amberlite XAD-2 (Japan Organo Co., Ltd., Tokyo, Japan, 5 cm $\times$ 90 cm). The elution solvents were used in the order of 1 L of water, 2 L of 20% methanol-water, 5 L of 50% methanol-water, 5 L of methanol, and 1 L of ethyl acetate.

Then, the eluates were obtained as follows: water eluate (fennel seeds: 6.0 g), 20% methanol-water eluate (fennel seeds: 2.0 g), 50% methanol-water eluate (fennel seeds: 3.9 g), methanol eluate (fennel seeds: 0.5 g), and ethyl acetate eluate (fennel seeds: 0.3 g). The 50% methanol-water eluate of fennel seed was further separated sequentially using silica gel column chromatography or gel filtration. Seven compounds—**3**, **4**, **7–11**—were isolated from the 50% methanol-water eluate of fennel seeds.

### 2.4. Isolated New Compounds

**5,5′-diallyl-2,2′-diglucopyranosyl-3,3′-dimethoxy diphenyl ether (7).**
FAB-MS: *m/z* 665[M-H]$^-$.

$^1$H-NMR (CD$_3$OD, δppm): 3.23 (4H, br.d, J = 6.5 Hz), 3.37~3.50 (6H, m), 3.68 (2H, br.d, J = 11.5 Hz), 3.80 (6H, m), 383~3.88 (4H, m), 4.99 (1H, m), 5.02 (2H, m), 5.07(1H, m), 5.89 (1H, dt, J = 6.5, 17 Hz), 5.92 (1H, dt, J = 6.5, 17 Hz), 6.38 (2H, d, J = 2 Hz), 6.52 (2H, d, J = 2 Hz).

$^{13}$C-NMR (CD$_3$OD, δppm): 41.0 (×2), 61.5 (×2), 62.4 (×2), 71.3 (×2), 74.9 (×2), 78.0 (×2), 78.1 (×2), 102.6 (×2), 109.2 (×2), 111.6 (×2), 115.9 (×2), 136.7 (×2), 138.7 (×2), 151.6 (×2), 152.1 (×2).

### 2.5. In Vitro Inhibition Test of AGEs Generation

The inhibition of AGE formation was examined as detailed in a previous report [28].

The mixture of the sample (20 µL), which was adjusted to the concentrations 0.1 mol/L phosphate buffer solution (PBS) (pH 7.4) (500 mL), distilled water (180 µL), 40 mg/mL of Bovine serum albumin (BSA, Sigma Chemical Co., Ltd., St. Louis, MO, USA) (200 mL), and 2 mmol/L of glucose aqueous solution (100 µL), was stirred. We prepared two samples of the same concentrations to see the difference in incubation. Furthermore, as a blank (controlled trial), we used methanol instead of a sample. Each sample was incubated for 30 h at 60 °C (A) and 25 °C (B), respectively. After incubation, trichloroacetic acid (100 µL) was added to the mixture and stirred. Then, the mixtures were centrifuged at 4 °C at 15,000 rpm for 4 min. The precipitates (AGEs) were dissolved with 1 mL of 0.25 N sodium hydroxide water solution-PBS and 200 µL was poured into a white microplate. The AGEs-derived fluorescence was measured using a microplate reader TECAN F200 (Tecan Group Ltd., Männedorf, Switzerland), at an excitation wave-length of 360 nm and a fluorescent wave-length of 440 nm. The percentage inhibition of AGE generation was calculated as,

AGEs inhibition rate (%) = {(blank A − blank B) − (sample A − sample B)/(blank A − blank B)} × 100

### 2.6. Cell Culture

TIG-110 cells (JCRB-05423) are normal diploid fibroblasts isolated from the skin of a 33-year-old Japanese woman. TIG-110 cells were cultured in T-25 flasks using DMEM containing fetal bovine serum and antibiotics (antibiotic-antimicrobial agent mixture solution (100 × concentration), Nakalai Tesque, INC., Kyoto, Japan) as cell culture medium. After 2–3 days of culturing in an incubator (37 °C, 5% CO$_2$), the cells grew to 80% confluency in the flasks. The 80% confluent cells were washed with PBS(−) solution; then, Trypsin solution (TrypLETM Express, ThermoFisher, Waltham, MA, USA) was added and the mixture was left to stand in a CO$_2$ 5% incubator at 37 °C for 5–8 min. Then, the cells were detached by gently tapping the flasks, checked under a microscope, and collected by centrifugation. After the cell count was measured, the number of cells was adjusted to the specified number and plates were seeded or passaged. TIG-110 cells were used for experiments up to 15 passages.

### 2.7. Cell Viability

Spice seed extracts and isolates were co-cultured with TIG-110 cells for 48 h to examine the cytotoxicity of the various spice seed extracts and isolates. Twenty-four hours before the start of the test, TIG-110 cells were seeded into 96-well plates. To ensure a uniform seeding concentration, DMEM was used to adjust the cell count to $5.0 \times 10^4$ cells/100 µL. After 24 h, sample-DMEM cultures containing diluted thawed compound and control were prepared and added to the seeded wells in 100 µL increments. Sample-DMEM was diluted and dissolved for each sample to achieve a final concentration of 25 µg/mL (0.4% DMSO concentration). The control was prepared from DMEM mixed with 0.4% DMSO. After 48 h, cell viability was checked by the MTT method. Cell viability was calculated as follows:

Cell viability (%) = (absorbance of Sample − DMEM/absorbance of control) × 100

*2.8. Determination of Glyoxal Concentration*

The GO concentration was examined to establish a glycation induction model test method. To obtain a final concentration of 5 mM of GO, 40% glyoxal (Fujifilm Wako Pure Chemical Co., Tokyo, Japan) was diluted and dissolved in DMEM. This 5 mM GO-DMEM solution was further diluted to prepare four concentrations (5 mM, 2.5 mM, 1.25 mM. and 0.625 mM) of GO-DMEM. Cells were seeded in the same manner as indicated previously herein. The prepared GO-DMEM was added to the seeded wells in 100 μL increments. The control was DMEM mixed with 0.4% DMSO. After 24 and 48 h, cell viability was examined by the MTT method. Cell viability was calculated as follows.

$$\text{Cell viability (\%)} = (\text{absorbance of GO} - \text{DMEM}/\text{absorbance of control}) \times 100$$

*2.9. Assay of AGE Formation Inhibitory Effects in Glyoxal System*

Based on the preliminary test results, GO-DMEM was set to 1.25 mM, and the culture time was set to 48 h. Using these conditions, samples that did not show cytotoxicity were co-cultured. Cell seeding and preparation of GO-DMEM were performed in the same method described previously herein. Samples were also diluted and dissolved in GO-DMEM. The control was 0.4% DMSO mixed with DMEM. After 48 h, cell viability was checked by the MTT method. Cell viability was calculated as follows.

$$\text{Cell viability (\%)} = (\text{absorbance of GO-DMEM, absorbance of sample-GO-DMEM}/\text{absorbance of control}) \times 100$$

*2.10. CML Qualitative Analysis (Immunostaining Method)*

CML, a type of AGE, was qualitatively determined by immunostaining. Glyoxal and compounds were added to collagen-coated chamber slides and cultured for 48 h. Then, 4% paraformaldehyde-PBS was added to each compartment as a fixative for immunostaining. Similarly, 0.5% triton X-100-PBS was used as a permeabilization agent. In addition, 5% BSA-PBS was used as a blocking agent. After pretreatment, staining was performed with primary and secondary antibodies. For staining of CML, a type of AGE, an anti-AGE monoclonal antibody (carboxymethyl lysine, Trans Genic Inc., Tokyo, Japan, Clone No. 6D12) was used as the primary antibody. The secondary antibody was Goat Anti-Mouse IgG H&L (Alexa Fluor 594; Abcam, Cambridge, UK, ab150116). For the staining of Vimentin, an intracellular cytoskeleton component, an anti-Vimentin antibody ab73159 (Abcam, Cambridge, UK,) was used as the primary antibody. Goat Anti-Chicken IgY H&L (Alexa Fluor 488; Abcam, Cambridge, UK, ab150169) was used as a secondary antibody. For DAPI (cell nuclei) staining, NucBlue$^{\text{TM}}$ Fixed Cell Stain ReadyProbes$^{\text{TM}}$ reagent (Invitrogen) was used. Goat Anti-Chicken IgY H&L (Alexa Fluor 488; Abcam, Cambridge, UK, ab150169) was used as a secondary antibody. For DAPI (cell nuclei) staining, NucBlue$^{\text{TM}}$ Fixed Cell Stain ReadyProbes$^{\text{TM}}$ reagent (Invitrogen, MA, USA) was used. After staining, chamber slides were checked using a fluorescence microscope CKX53 (Olympus Corporation, Tokyo, Japan).

*2.11. CML Quantitative Assay (ELISA)*

CML, a type of AGE, was quantified by ELISA. TIG-110 cells were seeded in 6-well plates 24 h before the start of the test. The cell count was adjusted to $1.0 \times 10^6$ cells/100 μL using DMEM to ensure a uniform seeding concentration. After 24 h, sample-DMEM, sample-GO-DMEM, and the control were prepared by diluting and dissolving the compound in DMEM and adding 2.0 mL each to the seeded wells. Sample-DMEM was diluted and dissolved until the final concentration of each sample reached 25 μg/mL (0.4% DMSO concentration). The control was DMEM mixed with 0.4% DMSO. After 48 h, the wells were washed twice with PBS to collect protein lysates, and 2.0 mL of distilled water was added to each well. The lysate was stored at $-80$ °C and dissolved immediately before CML determination. The lysate was tapped with distilled water and added to an Eppendorf tube, and the concentrated lysate was prepared by centrifugation. CML quantification was

performed following the protocol of the ELISA kit (Funakoshi Co., Ltd., Tokyo, Japan, STA-817, Lot. 10101451). Sample-CML expression levels were quantified using the CML-BSA standard provided in the kit.

### 2.12. Statistical Processing

Statistical processing was performed using SAS University Edition (SAS Institute, Cary, NC, USA.) with data expressed as the mean ± S.D. A risk rate of less than 5% (* $p < 0.05$, ** $p < 0.01$) was considered a significant difference.

## 3. Results

### 3.1. Coriander and Fennel Seeds Ingredients

Extracts of coriander (CSL) seeds were prepared in 70% methanol, and the crude extracts were extracted with 1-hexane and 1-butanol. The 1-butanol extracts showed the same level of AGEs inhibitory activity as that of aminoguanidine at the same concentration (Figure 1), and were therefore further purified by silica gel column chromatography and gel filtration.

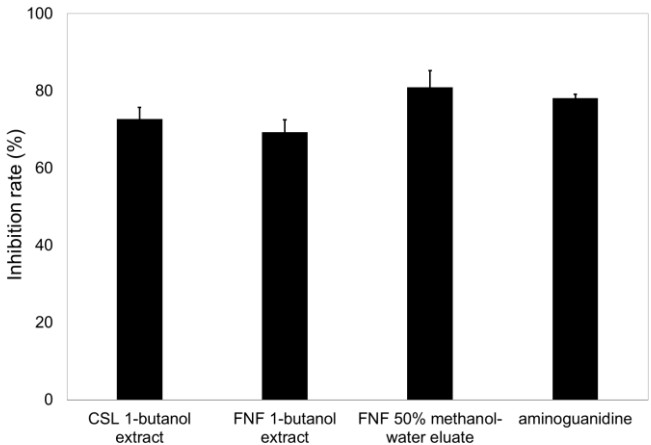

**Figure 1.** AGEs inhibitory activity of extracts. $n = 3$. Sample concentrations: 50 µg/mL. Aminoguanidine: positive control; CSL: Coriander; FNF: Fennel.

As a result, six compounds (Figure 2) were successfully isolated and identified as known compounds from MS and NMR spectra: hexyl-β-glucopyranoside (**1**) [29], chlorogenic acid (**2**) [30], quercetin 3-*O*-glucopyranoside (**3**) [31], (2*E*,6*R*)-2,6-dimethyl-2,7-octadien-6-ol-1-*O*-β-glucopyranoside (**4**) [32], adenosine (**5**) [29], and dehydroconiferyl alcohol-4-β-glucopyranoside (**6**) [33].

Extracts of fennel (FNF) seeds were prepared in 70% ethanol. The extracts were then extracted with 1-hexane and 1-butanol, and the 1-butanol extract was fractionated using Amberlite XAD-2. The 50% methanol eluate showed the same level of AGEs inhibitory activity as that of aminoguanidine at the same concentration (Figure 1) and was therefore further purified by silica gel column chromatography and gel filtration.

The following compounds were isolated and identified as known compounds from MS and NMR spectra: quercetin 3-*O*-glucopyranoside (**3**), (2*E*, 6*R*)-2,6-dimethyl-2,7-octadien-6-ol-1-*O*-β-D-glucopyranoside (**4**), syringin (**7**) [33], quercetin 4′-*O*-glucopyranoside (**9**) [34], kaempferol 4′-*O*-glucopyranoside (**10**) [34], and 6,8,4′-trimethoxy-5,7-dihydroxyl flavone (**11**) [34]. Furthermore, the novel compound 5,5′-diallyl-2,2′-diglucopyranosyl-3,3′-dimethoxy diphenyl ether (**8**) was found (Figure 2).

In this study, compound **8** was isolated from the 50% MeOH-H$_2$O eluate of fennel seeds as a new compound. The molecular formula of **8** was found to be C$_{32}$H$_{42}$O$_{15}$ by FAB-MS, which showed a characteristic peak at *m/z* 665 [M–H]$^-$, and the molecular weight was 666. The $^1$H-NMR spectrum of compound **8** indicated the presence of four aromatic signals, two methoxy groups, signals derived from two allyl groups, and glucoses. Furthermore, the

$^{13}$C-NMR spectrum of compound **8** was very similar to the known citrusin C spectrum [33], suggesting that it is a dimer of citrusin C, since a carbon signal was observed at δ 151.6 ppm. Based on this evidence, the structure of compound **8** was determined to be 5,5′-diallyl-2,2′-diglucopyranosyl-3,3′-dimethoxy diphenyl ether (Figure 2).

**Figure 2.** Isolated compounds.

*3.2. AGEs Inhibitory Activity*

The IC50 of the AGE inhibitory activity test for the nine compounds from which sufficient quantities of isolates were obtained is shown in Table 1. Since AGEs are formed quickly, we considered that a 30 h culture time would be efficient for testing the AGEs inhibitory activity of D-glucose and BSA. Aminoguanidine, which is known to have AGE-formation inhibitory activity, was used as a positive control. The $IC_{50}$ value of aminoguanidine, the positive control, was 0.42 mM; the $IC_{50}$ values of some of the isolates used in this study were lower than those of the positive control, indicating that they had high AGEs inhibitory activity.

**Table 1.** $IC_{50}$ values of the AGEs inhibitory activity test of the isolated compounds.

| Compound | AGEs Inhibitory Activity $IC_{50}$ Values |
|---|---|
| Aminoguanidine (positive control) | 0.42 |
| 1 | >1.0 |
| 2 | 0.21 |
| 3 | 0.10 |
| 4 | 0.35 |
| 7 | 0.28 |
| 8 | 0.08 |
| 9 | 0.08 |
| 10 | 0.10 |
| 11 | 0.18 |

Unit: mM.

Compound **8**, a new compound, demonstrated very high AGEs inhibitory activity with an $IC_{50}$ value of 0.08 mM.

### 3.3. TIG-110 Cell Viability (Cytotoxicity Test)

Extracts and isolates of each spice seed showed no cytotoxicity at a 25 μg/mL concentration (Figures 3 and 4), and there were no problems with their application to food and cosmetics.

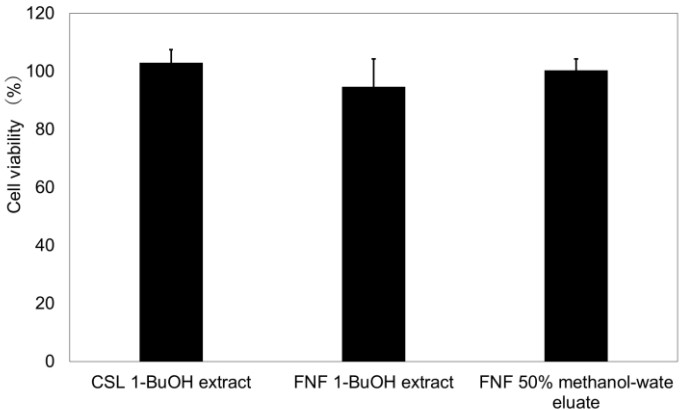

**Figure 3.** Cytotoxicity of extracts. $n = 3$. Sample concentration: 25 μg/mL; CSL: Coriander; FN: Fennel. The bars represent the mean + SD.

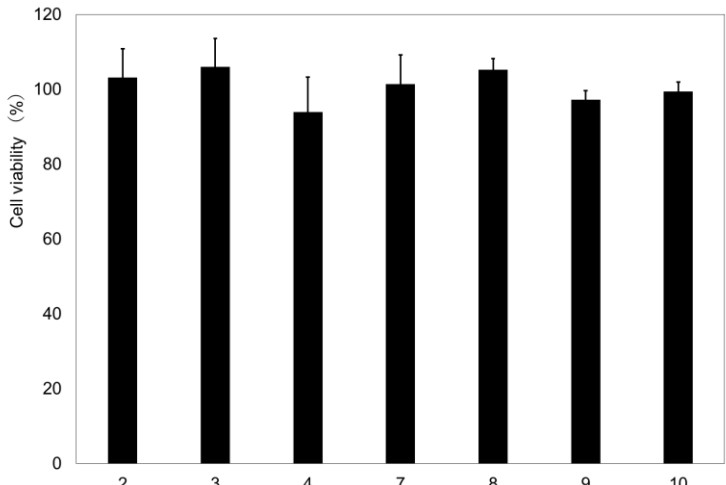

**Figure 4.** Compound cytotoxicity of isolated compounds. $n = 3$. Sample concentration: 25 μg/mL; compound **2**: chlorogenic acid, **3**: querucetin 3-*O*-glucopyranoside, **4**: (2*E*, 6*R*)-2, 6-dimethyl-2, 7-octadien-6-ol-1-*O*-β-D-glucopyranoside, **7**: syringin, **8**: 5,5′-diallyl-2,2′-diglucopyranosyl-3,3′-dimethoxy diphenyl ether, **9**: querucetin 4′-*O*-glucopyranoside, **10**: kaempferol 4′-O-glucopyranoside. The bars represent the mean + SD.

### 3.4. Preliminary Study in a Glycation-Induced Model

We established a model test method for glycation induction by exposing TIG-110 cells to GO. Aminoguanidine, which has anti-glycation activity, was used as a positive control. At 24 h (Figure 5) and 48 h (Figure 6) of culture, GO resulted in a concentration-dependent reduction in cell viability. At 24 h of culture, a concentration of 2.5 mM of GO significantly inhibited the reduction in cell viability compared to that with aminoguanidine (concentration of GO: 70.8 ± 7.8; concentration of GO: 2.5 mM + aminoguanidine: 89.4 ± 4.3). We also confirmed that GO concentrations of 2.5 mM and 1.25 mM significantly inhibited the reduction in cell viability compared to aminoguanidine at 48 h of culture (GO concentration 2.5 mM: 38.8 ± 10.0; GO concentration 2.5 mM + aminoguanidine: 94.0 ± 4.9; GO concentration 1.25 mM: 74.2 ± 8.0; concentration of GO 2.5 mM + aminoguanidine: 98.9 ± 8.0). These results indicated that the most stable and reproducible conditions were

48 h of culture and a GO concentration of 1.25 mM. We decided to conduct the following experiments using this model as a test method for the induction of glycosylation.

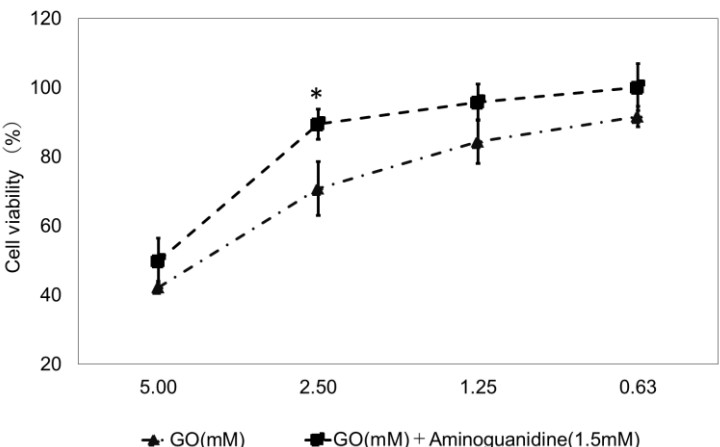

**Figure 5.** Cell viability of cells exposed to glyoxal (GO) at 24 h of culture. A total of four GO concentrations of 5 mM, 2.5 mM, 1.25 mM, and 0.625 mM were studied, and their effect on cell viability is shown. $n = 3$, GO: Glyoxal (mM); aminoguanidine: positive control (1.5 mM). The bars represent the mean + SD; * $p < 0.05$ versus GO; student *t*-test.

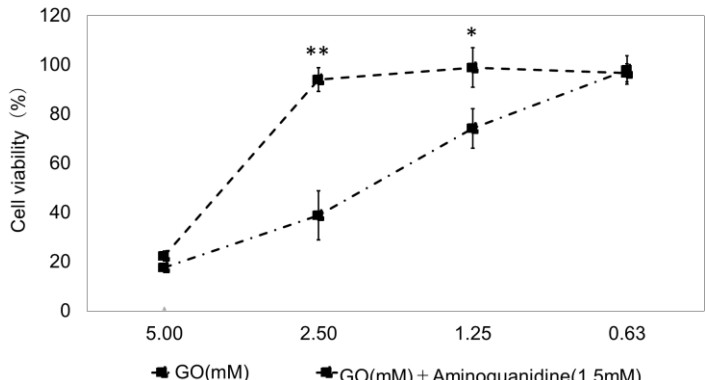

**Figure 6.** Cell viability of cells exposed to glyoxal (GO) at 48 h of culture. Four glyoxal (GO) concentrations of 5 mM, 2.5 mM, 1.25 mM, and 0.625 mM were studied, and their effects on cell viability are shown. $n = 3$. GO: Glyoxal (mM); aminoguanidine: positive control (1.5 mM). The bars represent the mean + SD; * $p < 0.05$; ** $p < 0.01$ versus GO; student *t*-test.

*3.5. TIG-100 Cell Viability and Effect of the Spice Seed on Glyoxal Activity*

In a glycation-induced model study, each spice extract was evaluated at 25 μg/mL without cytotoxicity (Figure 7). The 1-BuOH extracts of all spice seed significantly reduced cell viability (GO concentration 1.25 mM: 71.1 ± 4.8; GO concentration 1.25 mM + CSL 1-BuOH extract: 95.7 ± 5.2; GO concentration 1.25 mM + FNF 1-BuOH extract: 101.3 ± 5.9).

To investigate the contribution ingredient of each extract, we evaluated the isolated compounds (Figure 8). As a result, as shown in the figure, we found a total of four compounds—**2**, **4**, **8**, and **10**—that resulted in a significant improvement in reducing cell viability (GO concentration 1.25 mM: 71.1 ± 4.8; GO concentration 1.25 mM + **2**: 100.8 ± 7.7; GO concentration 1.25 mM + **4**: 99.3 ± 3.5; GO concentration 1.25 mM + **8**: 99.8 ± 5.7; GO concentration 1.25 mM + **10**: 99.7 ± 5.4).

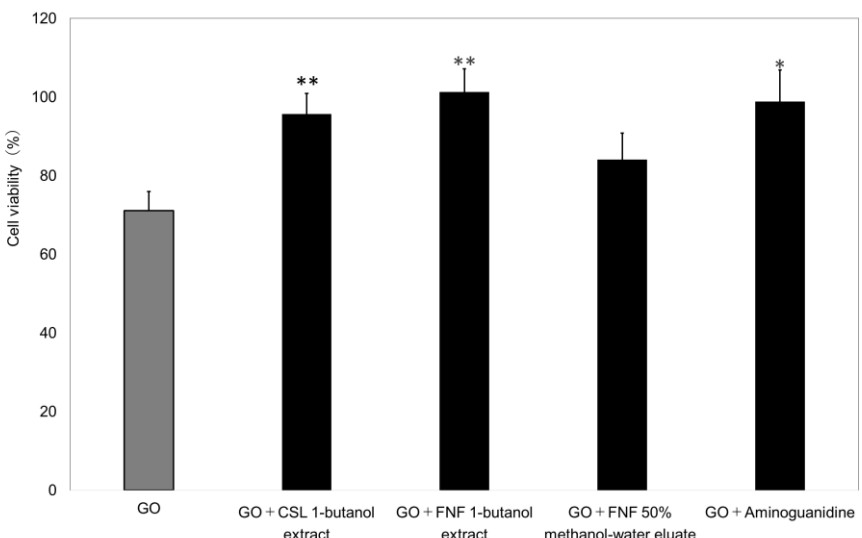

**Figure 7.** Glycation suppression experiment with selected extracts in the presence of glyoxal (GO) using TIG-110 cells. *n* = 3, GO: Glyoxal (1.25 mM); aminoguanidine: Positive control (1.5 mM); samples (CSL: Coriander, FNF: Fennel); sample concentration: 25 μg/mL. The bars represent the mean + SD, * *p* < 0.05, ** *p* < 0.01 versus GO; student *t*-test.

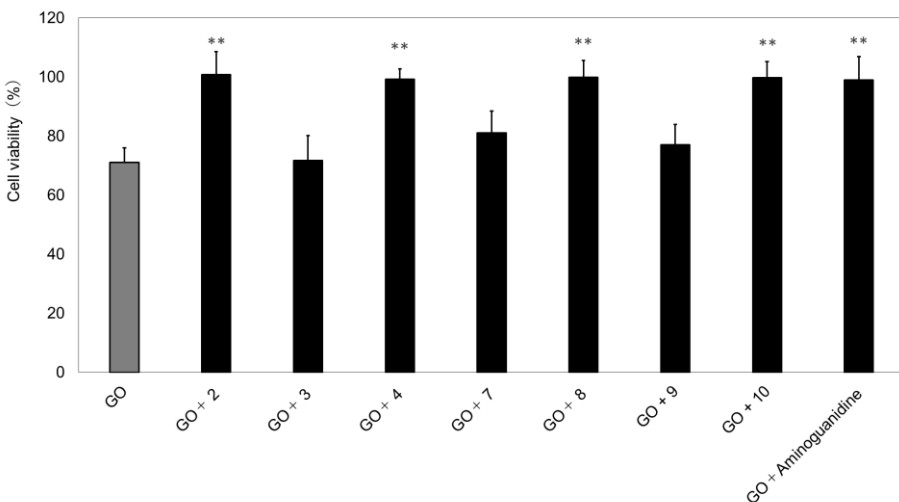

**Figure 8.** Glycation suppression with selected purified compounds in the presence of glyoxal (GO) in TIG-110 cells. *n* = 3. GO: Glyoxal (1.25 mM); aminoguanidine: Positive control (1.5 mM;, compounds **2**: chlorogenic acid, **3**: querucetin 3-*O*-glucopyranoside, **4**: (2*E*, 6*R*)-2, 6-dimethyl-2,7-octadien-6-ol-1-*O*-β-D-glucopyranoside, **7**: syringin, **8**: 5,5′-diallyl-2,2′-diglucopyranosyl-3,3′-dimethoxy diphenyl ether, **9**: querucetin 4′-*O*-glucopyranoside, **10**: kaempferol 4′-O-glucopyranoside); sample concentration: 25 μg/mL (0.4% DMSO. The bars represent the mean + SD; ** *p* < 0.01 versus GO; student *t*-test.

### 3.6. CML Qualitative Analysis (Immunostaining Method) of the Isolated Compound

The reduction in cell viability was confirmed by the glycation-induced model test method. We therefore sought to visualize the intracellular or extracellular matrix glycation induced by exposure to GO. As shown in Figure 9, CML (red), Vimentin (green), and DAPI (blue) were subjected to immunostaining. The exposure of TIG-110 cells to GO stimulated the formation of CML. The addition of the compound also inhibited the formation of CML.

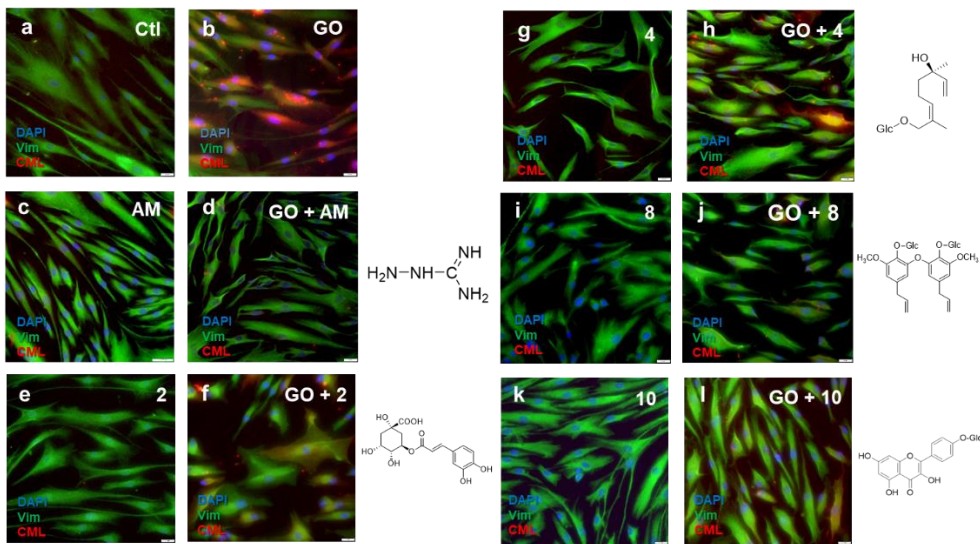

**Figure 9.** Images of immunofluorescent-stained glycation end products (CML) with the isolated compounds. CML (red); Vimentin (green); DAPI (blue); Ctl: control (**a,b**); aminoguanidine: 1.5 mM (**c,d**); compounds **2**: chlorogenic acid (**e,f**), **4**: (2*E*, 6*R*)-2, 6-dimethyl-2,7-octadien-6-ol-1-*O*-β-D-glucopyranoside (**g,h**), **8**: 5,5′-diallyl-2,2′-diglucopyranosyl-3,3′-dimethoxy diphenyl ether (**i,j**), **10**: kaempferol 4′-*O*-glucopyranoside (**k,l**); GO: 1.25 mM (**b,d,f,h,j,l**); sample concentration: 25 μg/mL.

### 3.7. CML Quantitative Assay (ELISA) of the Isolated Compound

The immunostaining method confirmed that GO exposure induced the glycation reaction. Hence, we performed a quantitative determination of CML by ELISA and a qualitative immunostaining method. As shown in Figure 10, ELISA showed the production of CML. Culturing GO and TIG-110 cells for 48 h stimulated the production of CML. The addition of compounds also inhibited the generation of CML (GO: $0.59 \pm 0.02$; GO + **4**: $0.39 \pm 0.09$; GO + **8**: $0.46 \pm 0.03$; GO + aminoguanidine: $0.27 \pm 0.08$).

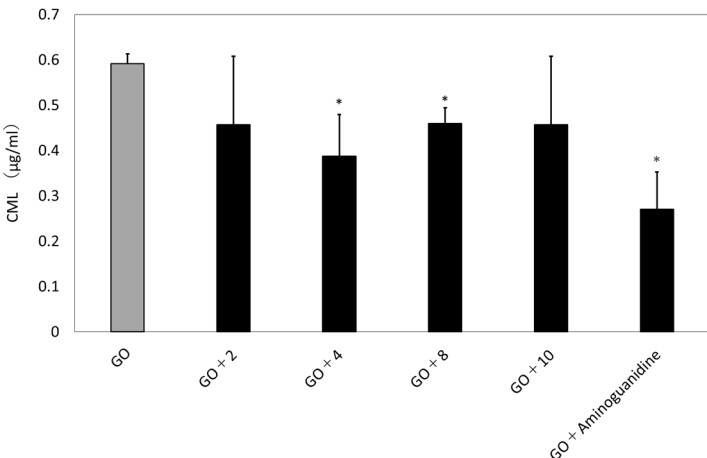

**Figure 10.** Amount of glycation end product (CML) formation with isolated compounds. *n* = 3. GO: Glyoxal (1.25 mM); aminoguanidine: positive control (1.5 mM); compounds **2**: chlorogenic acid, **4**: (2*E*, 6*R*)-2, 6-dimethyl-2,7-octadien-6-ol-1-*O*-β-D-glucopyranoside, 8: 5, 5′-diallyl-2,2′-diglucopyranosyl-3, 3′-dimethoxy diphenyl ether, 10: kaempferol 4′-*O*-glucopyranoside; sample concentration: 25 μg/mL. The bars represent the mean + SD; * $p < 0.05$ versus GO; student *t*-test.

## 4. Discussion

Glycation continues to react very slowly in vitro, and AGEs are formed as the final form.

Such AGEs cause a loss of original biological functions. AGEs have been identified in the lysine and arginine residues of collagen fibers that contribute to skin elasticity in humans. Previous studies have also identified AGEs in human skin and reported that their quantity increases in an age-dependent manner [14]. In addition to AGEs in skin, the presence of AGEs in the collagen of blood vessel walls has been identified. These are considered to increase the susceptibility to aging-related diseases, such as atherosclerosis [35]. AGEs can be considered the cause of these disorders and removing these AGEs can inhibit the progression of pathological aging [36], making them targets for aging prevention.

AGEs have been reported to have a variety of structures and structurally complex AGEs have not yet been elucidated. However, some AGEs have been identified in general glycation reactions, which are non-enzymatic reactions between reducing sugars and proteins. Among the AGEs with known structures, those with cross-linked networks are mainly fluorescent substances, which are detectable. In recent years, using this fluorescence feature, substances and compounds that suppress AGE formation have been actively explored.

Our anti-glycation activity test, as described, was conducted to evaluate the inhibitory effect on AGE formation by inducing the formation of AGEs in vitro. These AGEs were generated using reducing sugars and BSA in an experimental system. This testing system is widely used to evaluate the anti-glycation activity of plant extracts and fractions in the purification process because the glycation reaction occurs in a short time [37]. In this study, we evaluated the anti-glycation effect of the compounds as an experimental system using the reducing sugar D-glucose; the culture time was also examined. In a similar experimental system, Sato et al. (2017) [25] performed an evaluation of anti-glycation activity using D-ribose as the reducing sugar. In the D-ribose experimental system, the $IC_{50}$ value of aminoguanidine, the positive control, was 0.27 mM. The experimental system that we used in this study was an anti-glycation activity test using D-glucose, and the $IC_{50}$ value of aminoguanidine, the positive control, was 0.42 mM. These are both reducing sugars, specifically D-ribose and D-glucose, and the $IC_{50}$ values are not considered significantly different. In this study, we employed a test system using D-glucose, which commonly circulates in human blood as blood glucose, as the anti-glycation activity test. It is believed that this test system can efficiently evaluate many samples. Using this activity test as a benchmark, we searched for components in two spices. We found that the 1-BuOH extracts of coriander and fennel seeds had exceptionally high AGEs inhibitory activity. The extracts were separated by silica gel column chromatography and gel filtration to examine the contribution ingredient of the activity. In total, 11 compounds were isolated from two spice seeds. Several isolated compounds showed lower concentrations than the $IC_{50}$ value of aminoguanidine, the positive control, and showed high AGEs inhibitory activity.

Furthermore, in this study, we mainly focused on establishing experimental conditions for the glycation induction model test method. We evaluated spice seed extracts and compounds obtained in the exploration using this test model. In this model study, GO, a type of aldehyde, was used as a dicarbonyl compound produced by the autoxidation of glucose [38]. This dicarbonyl compound is a glycation intermediate and is highly reactive with proteins, efficiently generating CML, a type of AGE. Based on these facts, we decided to examine a model test for the induction of glycation by adding GO to cells for culturing.

We evaluated 11 compounds obtained from spice seeds, 9 of which were available in sufficient quantities for evaluation. Evaluation results using this model test indicated that the following four compounds significantly reduced cell viability in the presence of glycosylation: chlorogenic acid (**2**), (2*E*,6*R*)-2,6-dimethyl-2,7-octadien-6-ol-1-*O*-β-D-glucopyranoside (**4**), 5,5′-diallyl-2,2′-diglucopyranosyl-3,3′-dimethoxy diphenyl ether (**8**), and kaempferol 4′-*O*-glucopyranoside (**10**).

This model test confirmed the inhibition of cell viability via the addition of the compound under glycation induction conditions. Furthermore, we investigated CML based on qualitative and quantitative determination to examine the formation of intracellular CML. CML formation was inhibited by adding these compounds based on results of the

qualitative immunostaining method. Five compounds, including aminoguanidine, significantly inhibited the formation of CML in the ELISA method for quantitative analysis. These results suggest that apoptosis and the generation of CML are closely related.

Aminoguanidine, a positive control, has a known mechanism of action with respect to carbonyl compound inhibition. The amino group of aminoguanidine traps GO, significantly reducing its reactivity and inhibiting CML formation [39]. The two compounds isolated in this study that inhibited the formation of CML were (2E,6R)-2,6-dimethyl-2,7-octadien-6-ol-1-*O*-β-D-glucopyranoside (**4**), and 5,5′-diallyl-2,2′-diglucopyranosyl-3,3′-dimethoxy diphenyl ether (**8**). These two compounds are structurally characterized by the presence of a double bond at the end of each compound. This terminal double bond likely inhibits the formation of CML by trapping the aldehyde group of GO.

Some compounds did not inhibit CML formation, but did inhibit apoptosis. There is a pathway involving CML that leads to apoptosis. The compounds chlorogenic acid (**2**) and kaempferol 4′-O-glucopyranoside (**11**) had no inhibitory effect on CML formation in this study, but might inhibit the CML-derived apoptotic pathway. Previously, Alikhani et al. (2005) [40] reported that CML activates caspase-3, a factor involved in fibroblast apoptosis. Next, caspase-8 and caspase-9, which exist downstream of caspase-3 activity, were found to be activated as well [41]. We believe that these two compounds inhibit the caspase-3, 8, and 9 pathways, which are factors in apoptosis. Therefore, it is essential to examine whether these compounds inhibit this activity and investigate the structural characteristics of these compounds.

We confirmed that GO used in this study is a versatile agent that induces glycation, leading to the formation and accumulation of CML. This model test proved effective because the glycation reaction in vitro was very relaxed and immediate over 48 h.

However, whereas there are differences between this model test and the glycation reaction in vitro, it is difficult to identify these differences. Thus, we believe that this model test can efficiently evaluate the anti-glycation effects of compounds and help to obtain primary data.

## 5. Conclusions

We isolated 11 compounds from two spice seeds, namely coriander and fennel, and found several substances that showed anti-glycation activity. A new compound (5,5′-diallyl-2,2′-diglucopyranosyl-3,3′-dimethoxy diphenyl ether) was isolated from fennel seeds and showed high anti-glycation activity with an IC50 value of 0.08 mM, thereby indicating a high anti-glycosylation activity. In this study, we established a glyoxal (GO)-induced glycation test method for human skin cells, confirmed the anti-glycation effect of spice seeds using this glycation induction model, and found that the exposure of TIG-110 human skin-derived fibroblast cells to GO reduced cell viability. The most stable conditions for cell viability were found to be a GO concentration of 1.25 mM and a culture time of 48 h. We evaluated extracts and isolates of spice seeds using this model as a model test for glycation induction. The results showed that the 1-BuOH extract of each spice seed and a total of four isolated substances (chlorogenic acid (**2**), (2*E*, 6*R*)-2, 6-dimethyl-2,7-octadien-6-ol-1-*O*-β-D-glucopyranoside (**4**), 5,5′-diallyl-2,2′-diglucopyranosyl-3,3′-dimethoxy diphenyl ether (**8**), and kaempferol 4′-*O*-glucopyranoside (**10**)) significantly inhibited the decrease in cell viability. We conducted qualitative and quantitative analyses of carboxymethyl lysine (CML), a type of AGE, to determine the relationship between cell viability and AGEs. The relationship between cell viability and the amount of CML was correlated. Establishing a glycation induction model test using skin cells makes it possible to quickly screen extracts of natural ingredients in the future. We conclude that the extracts and isolated compounds obtained in this study may be effectively used for food additives and cosmetic materials as AGE generation inhibitors.

**Author Contributions:** A.S. and R.T. conceived and designed the research. A.Y. carried out all experiments. M.F. carried out the isolation of functional ingredients in fennel seeds. A.S. and A.Y. were responsible for writing—review and editing. All authors have read and agreed to the published version of the manuscript.

**Funding:** A part of this research was funded by the 21st Century Joint Research Enhancement Grant from Kindai University, grant number KD201705 and KD2003.

**Institutional Review Board Statement:** Not applicable.

**Informed Consent Statement:** Not applicable.

**Data Availability Statement:** The data presented in this study are available on request from the corresponding author. The data are not publicly available due to privacy.

**Acknowledgments:** We would also like to extend my gratitude to Marie Tomojiri, Ayaka Tsukanishi, and Masato Azuma, who were undergraduate students when they were part of the research team, for their participation in the project.

**Conflicts of Interest:** The authors declare no conflict of interest.

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
