# Peer review of "Development of Evaluation Methods for Anti-Glycation Activity and Functional Ingredients Contained in Coriander and Fennel Seeds"

_processes, doi:10.3390/pr10050982_

Round 1

Reviewer 1 Report

Dear Authors, I have reviewed the MS entitled Development of Evaluation Methods for Anti-glycation Activity and their Functional Ingredients Contained in Coriander, and Fennel Seeds and suggest the following:

-in general, please use passive voice

-in general, please remove comma before "and"

-line 46, please remove space between Perilla and comma

-line 46-47, please format, according to journal requirements, writing of Latin names (italics?)

-line 35: are there 2 or 3 seeds taken into account? Please correct

-line 62, please correct word all thre herbs

-line 153, please correct word mrthanol

-line 214, please remove second )

-line 265, please add space between -80 and oC

-line 287, please add space between (3) and [31]

-line 299, please remove the second space between of and AGEs in same level of AGEs inhibitory

-is querucetin correct? or quercetin?

Author Response

To Reviewers

              Thank you very much for valuable suggestions and comments to our manuscript.   According to the suggestions of the reviewer, we prepared a revised manuscript.   Followings are major points of revision and our comment.

              We agree with the suggestion of reviewer.   Therefore, We changed everything according to the suggestion of reviewer.

              Because we got a wrong compound number of figure 2, we corrected a figure.

1. -in general, please use passive voice

According to suggestion, we revised the parts of manuscript.

2. -in general, please remove comma before "and"

According to suggestion, we corrected it.

3. -line 46, please remove space between Perilla and comma 

According to suggestion, we corrected it.

4. -line 46-47, please format, according to journal requirements, writing of Latin names (italics?)

According to suggestion, we corrected it.

5. -line 35: are there 2 or 3 seeds taken into account?

According to suggestion, we corrected it to ‘two seeds’.

6. -line 62, please correct word all thre herbs 

According to suggestion, we corrected it to ‘all the herbs’.

7. -line 153, please correct word mrthanol

According to suggestion, we corrected it to ‘methanol’.

8. -line 214, please remove second )

According to suggestion, we corrected it.

9. -line 265, please add space between -80 and °C

According to suggestion, we corrected it.

10. -line 299, please remove the second space between of and AGEs in same level of AGEs inhibitory

According to suggestion, we corrected it.

11. -line 214, please remove second )

According to suggestion, we corrected it.

12. -is querucetin correct? or quercetin?

According to suggestion, we corrected it to ‘quercetin’.

Reviewer 2 Report

The manuscript Development of Evaluation Methods for Anti-glycation Activity and their Functional Ingredients Contained in Coriander, and Fennel Seeds is a very interesting work, with many methods of analysis, with a captivating approach. The working methods were very well explained, but I would have a few suggestions:

-The abstract seems to me far too long, very detailed. I think it should be more succinct

-L62 "thre" means "the"?

-L92-95 Why did you choose this method?

-L128-132 Why did you choose to use different solvents?

-L214 you put two brackets, delete one

-L18 You noted that two spice seeds were used in this study, then at L541-542 another species appeared that are not given in the manuscript. I believe that the Conclusions need to be restored and that they should be dealt with in more detail.

Good luck.

Author Response

To Reviewers

              Thank you very much for valuable suggestions and comments to our manuscript.   According to the suggestions of the reviewer, we prepared a revised manuscript.   Followings are major points of revision and our comment.

              We agree with the suggestion of reviewer.   Therefore, We changed everything according to the suggestion of reviewer.

              Because we got a wrong compound number of figure 2, we corrected a figure.

-The abstract seems to me far too long, very detailed. I think it should be more succinct

According to suggestion, we corrected abstract briefly.

-L62 "thre" means "the"?

According to suggestion, we corrected it to ‘the’.

-L92-95 Why did you choose this method?

Because a lot of CML is present in skin, we examined this method.

-L128-132 Why did you choose to use different solvents?

In preliminary experiment of the extraction, we chose a solvent with much quantity of the extract.

-L214 you put two brackets, delete one

According to suggestion, we corrected it.

-L18 You noted that two spice seeds were used in this study, then at L541-542 another species appeared that are not given in the manuscript. I believe that the Conclusions need to be restored and that they should be dealt with in more detail.

According to suggestion, we corrected conclusions.

Round 2

Reviewer 2 Report

It seems ok to me, thanks for considering the suggestions.